# Iatrogenic Hallux Varus in a Patient with Rheumatoid Arthritis

**DOI:** 10.3390/healthcare13030217

**Published:** 2025-01-21

**Authors:** Mercedes Ortiz-Romero, Alvaro Fernandez-Garzon, Manuel Pabon-Carrasco, Aurora Castro-Mendez, Luis M. Gordillo-Fernandez

**Affiliations:** 1Department of Podiatry, Faculty of Nursing, Physiotherapy and Podiatry, University of Seville, 41009 Seville, Spain; mortiz17@us.es (M.O.-R.); auroracastro@us.es (A.C.-M.); lgordillo@us.es (L.M.G.-F.); 2Private Practice in Podocorp Clinic, 11201 Algeciras, Spain; 3“CTS-1054: Interventions and Health Care, Red Cross (ICSCRE)”, Nursing Department, Faculty of Nursing, Physiotherapy and Podiatry, University of Seville, 6 Avenzoar ST, 41009 Seville, Spain; mpabon2@us.es

**Keywords:** hallux varus, rheumatoid arthritis, arthrodesis, foot deformity, surgical treatment, joint preservation

## Abstract

**Background/Objectives:** Iatrogenic hallux varus is a rare complication often arising after hallux valgus surgery, characterized by medial deviation of the hallux. This report presents the case of a 58-year-old female with iatrogenic hallux varus complicated by rheumatoid arthritis (RA). The objective is to highlight the challenges and outcomes of surgical treatment in RA patients with complex foot deformities. **Methods:** The patient presented with severe medial deviation of the hallux and claw positioning of the lesser toes, resulting in pain and functional limitations. Radiological analysis indicated overcorrection of the first intermetatarsal angle and deformity of the lesser toes. Surgical management included arthrodesis of the first metatarsophalangeal (MTP) joint using K-wires and resection arthroplasty of the lesser metatarsals. **Results:** Postoperative outcomes revealed correct alignment, pain reduction, and restoration of functional capabilities. However, a non-union was observed in the first MTP arthrodesis after 24 months, which remained asymptomatic. **Conclusions:** This case underscores the importance of careful surgical planning in RA patients to balance joint preservation and deformity correction. Arthrodesis proved effective for stability and pain relief in RA-associated deformities, although long-term follow-up remains critical to address complications. Tailored interventions are necessary to improve the quality of life in RA patients with complex foot deformities.

## 1. Introduction

Rheumatoid arthritis (RA) is a chronic autoimmune disease that primarily affects the synovial joints, leading to symmetric polyarthritis and progressive musculoskeletal deformities [1]. Foot involvement is highly prevalent, occurring in approximately 90% of RA patients, with deformities such as hallux valgus reported in up to 54% of cases, significantly impacting patients’ quality of life due to pain, instability, and functional limitations [2].

Hallux varus is an uncommon deformity defined as the medial deviation of the hallux at the metatarsophalangeal (MTP) joint [3]. Radiologically, it is characterized by a negative MTP joint angle. Although rare, hallux varus can result from iatrogenic causes, particularly overcorrection during hallux valgus surgery, with reported incidences ranging from 2% to 15% [4]. The deformity can also arise due to traumatic, post-surgical, inflammatory, or congenital factors. The primary pathophysiology involves an imbalance of muscle-tendon forces, notably between the flexor hallucis brevis and the sesamoid ligaments [5]. Excessive soft tissue release during surgery, including lateral capsule and sesamoid excision, exacerbates this imbalance [6].

Hallux varus presents as a cosmetic deformity or difficulty in shoe fitting, with less frequent complaints of pain [7]. Clinical evaluation focuses on the flexibility of the deformity and associated issues, such as clawing of the lesser toes or compensatory hindfoot supination [8]. Radiographic analysis confirms the diagnosis and evaluates the deformities, commonly identifying a zero or negative MTP angle along with additional structural abnormalities, such as arthritic changes or length discrepancies [9].

This case is both novel and significant as it describes the successful surgical management of a severe iatrogenic hallux varus deformity in a patient with rheumatoid arthritis (RA). The approach combined MTP joint arthrodesis with resection arthroplasty of the lesser metatarsals. It underscores the importance of customized surgical planning and long-term follow-up in addressing complex foot deformities in RA patients, a topic that is rarely discussed in the existing literature.

## 2. Case Report

We present the case of a 58-year-old female patient. The patient has a history of circulatory problems, a history of rheumatoid arthritis and a previous operation for Tailor’s bunion and hallux valgus.

Clinical examination reveals a varus deviation of the first toe and claw position of the lesser toes (Figure 1). Manual reduction of the varus is not possible, and movement of the hallux is painful. In addition, the position of the hallux prevents the patient from putting on shoes normally.

Finally, the radiological study aids in assessing the degree of deformity and the condition of the joints (Figure 2). It also provides insight into the surgical techniques previously employed. It is likely that her prior surgery involved a Silver exostectomy technique combined with soft tissue release, which contributed to the current condition of the toe. Evidence of a Silver exostectomy is also observed on the fifth metatarsal head.

The patient’s medical examination revealed no systemic, cardiac or neurological diseases.

The medical examination confirmed the presence of a rheumatologic disease, which likely contributed to the failure of the previous hallux valgus surgery.

The patient was diagnosed with rheumatoid arthritis at the age of 40 and is currently undergoing treatment with methotrexate.

Despite the ongoing treatment, the patient continues to experience significant pain in her right foot, leading her to opt for hallux valgus surgery. However, following the failure of this initial procedure, the pain worsened, necessitating a second surgery to address the iatrogenic hallux varus and associated rheumatoid pain.

Hallux varus results from an imbalance between the medial and lateral soft tissue forces of the first radius. Key factors include the excessive release of lateral tissues, such as the sesamoid ligaments and joint capsule, which decreases lateral stability. Simultaneously, the retraction of medial tissues, such as the adductor hallucis, can exacerbate medial deviation, particularly in patients with poor bone quality due to rheumatoid arthritis.

To minimize these complications, key surgical principles should be followed: Avoid excessive lateral soft tissue releases during corrective procedures. Perform strategic tendon transfers, such as adductor hallucis, to rebalance forces. Carefully repair the joint capsule while preserving its functionality. Implement postoperative monitoring and early rehabilitation to optimize recovery and prevent fibrosis.

These strategies, in combination with thorough preoperative evaluations and intraoperative adjustable techniques, can reduce the incidence of postoperative deformities such as hallux varus.

### 2.1. Surgical Procedure

The patient receives a spinal block with 0.75% bupivacaine, along with sedation administered by the anesthesiology team. A tourniquet is placed 10 cm below the fibular head with a pressure of 250 mm Hg.

First, we made longitudinal incisions on the dorsomedial aspect of the first MTP joint, on the third metatarsal and on the lateral aspect of the fifth metatarsal. In addition, we made semi-elliptical incisions on the proximal interphalangeal (IFP) joints of the second, third and fourth toes.

A pan-resection of the metatarsal heads of the second, third, fourth and fifth metatarsals was performed for the purpose of MTP joint arthroplasty. The articular surfaces of the first MTP joint were removed, and small holes were drilled in the head and base of the hallux to promote and facilitate bone regeneration. Additionally, the articular surfaces of the proximal interphalangeal (IFP) joints of the 2nd, 3rd, and 4th toes were removed to create stable bases for subsequent arthrodesis

Osteosynthesis was performed using K-wires (Figure 3). Three 2 mm K-wires were used for the triplanar fixation of the hallux arthrodesis, and 1.6 mm K-wires were used for the fixation of the IFP joints.

In addition to metatarsophalangeal (MTP) joint fusion, medial and lateral soft tissue balance was carefully evaluated during the second surgery. Intraoperative dynamic testing was conducted using temporary fixation with K-wires to verify passive hallux alignment and identify abnormal stress patterns. This approach enabled the selective release of the medial capsule to alleviate excessive tension while preserving the integrity of the lateral sesamoid ligaments. These measures ensured an even distribution of forces and optimized functional outcomes following the fusion procedure.

Sutures were performed using 2/0 synthetic absorbable sutures for the closure of the capsules of the first MTP and lesser rays, 3/0 sutures for the closure of the fascia and 4/0 Biosyn^TM^ monofilament sutures for continuous skin closure in all incisions. 

### 2.2. Postsurgical Procedure and Evolution

The patient was partially loaded from the first day with a post-surgical shoe for 8 weeks. Weekly dressings were carried out to monitor post-surgical progress and prevent infection.

At 3 weeks post-surgery, K-wires were removed from the lesser toes. The first radius K-wires remained in place until 8 weeks post-surgery under radiological follow-up. Once the K-wires had been removed, the patient began physiological weight-bearing in a sports shoe. After 12 weeks post-surgery, the patient had no pain or inflammation.

After 24 months post-surgery, a non-union of her first MTP was observed. The non-union was symptom-free. The patient is able to walk normally, has no pain, and has a foot that allows her to play sports and lead a normal life.

## 3. Results

Postoperative follow-up demonstrated significant improvements in the patient’s clinical condition. By 12 weeks post-surgery, the patient reported complete resolution of pain and returned to daily activities, including walking and wearing regular footwear. At 24 months, the patient remains asymptomatic, able to walk and participate in sports without limitations.

Biomechanical analysis conducted at the latest follow-up revealed a well-balanced distribution of plantar loads. The first MTP exhibited effective weight-bearing, contributing to enhanced stability during gait, while the central rays showed no evidence of overload. These findings, combined with the patient’s subjective reports of improved functionality and absence of pain, underscore the success of the surgical intervention in restoring biomechanical efficiency and optimizing foot performance.

The postoperative radiological (Figure 4) study showed the correct alignment of all the toes. A good evolution of the resection of metatarsal heads and an asymptomatic non-union of the first MTP joint was observed.

The patient is continuing her treatment for rheumatoid arthritis to maintain pain-free function in the remaining joints. Regarding the operated foot, the patient is now pain-free due to the removal of the affected joints.

## 4. Discussion

The management of foot deformities in patients with rheumatoid arthritis (RA) presents unique challenges due to the progressive nature of the disease and the structural damage it causes [10,11]. This case of a 58-year-old female patient with iatrogenic hallux varus highlights the complexity of treating deformities in the context of RA and previous surgical interventions [12]. The discussion explores the implications of the case, the surgical approach, and the broader context of treating hallux varus in RA patients, integrating evidence from the literature [13,14].

### 4.1. Clinical Implications of Hallux Varus in RA

This manifests as the medial deviation of the hallux, causing difficulties with shoe fitting and ambulation, as seen in the presented case [9].

The patient’s clinical presentation included a varus deviation of the hallux and clawing of the lesser toes. This deformity was rigid and painful, further complicating daily activities such as walking and wearing shoes. Radiological studies confirmed severe structural abnormalities, including overcorrection of the first intermetatarsal angle and deformities of the lesser toes [10]. Such findings underscore the importance of preoperative planning and thorough assessment in RA patients, as the risk of surgical complications is inherently higher due to systemic inflammation and reduced bone quality [11]

### 4.2. Surgical and Rationale

The primary goals of surgical intervention in RA-associated foot deformities are to reduce pain, restore alignment, and improve functionality [14]. In this case, arthrodesis of the first metatarsophalangeal (MTP) joint and resection arthroplasty of the lesser metatarsals were chosen as the surgical technique. Arthrodesis is widely regarded as the gold standard for correcting hallux varus, particularly in cases where joint degeneration or significant deformity is present [15]. The decision to combine arthroplasty reflects the need to address deformities in the lesser toes, which often contribute to forefoot pain and functional impairment [16].

The surgical procedure involved multiple incisions to access the affected joints and correct the deformities. The use of K-wires for osteosynthesis was dictated by the patient’s poor bone quality, a common challenge in RA patients [15]. K-wire fixation provides sufficient stability to accommodate the compromised structural integrity of the bones. While screw fixation is generally preferred for its superior mechanical stability, K-wires remain a viable option in cases where bone quality is inadequate [16,17].

The postoperative course for the patient was favorable, with correct alignment of the toes achieved and maintained over a 48-month follow-up period. The patient experienced significant pain relief, regained mobility, and was able to wear normal footwear. These outcomes align with findings in the literature, demonstrating that arthrodesis of the first MTP joint is effective in relieving pain and restoring functionality in RA patients [18,19].

However, the case also highlights a common complication of arthrodesis: during follow-up, a non-union was observed at the first MTP joint. Notably, this non-union was asymptomatic and did not compromise the patient’s functionality. This finding reflects the complex interplay between surgical technique, patient-specific factors, and long-term outcomes. While non-union is a recognized complication, its clinical significance varies [18]. In this case, the lack of symptoms suggests that the overall construct stability was sufficient to support the patient’s functional demands [19].

### 4.3. Broader Context of Hallux Varus Management

The literature on hallux varus emphasizes the importance of tailored surgical approaches based on the severity of the deformity, the presence of joint degeneration, and patient-specific factors such as bone quality and comorbidities [20]. Early-stage hallux varus can sometimes be managed conservatively with functional taping, splinting, and physical therapy [21]. However, such approaches are less effective in chronic, rigid deformities like the one presented in this case [22].

For RA patients, joint preservation techniques such as the Scarf osteotomy may be considered joint degeneration [20]. This technique has been shown to achieve high patient satisfaction rates and satisfactory correction of hallux valgus deformities [13]. However, for hallux varus associated with significant joint damage, arthrodesis remains the preferred option [18], offering long-term stability and pain relief while minimizing the risk of recurrence [19].

The choice of fixation technique is another critical consideration. While K-wires were used in this case due to poor bone quality, they are generally considered inferior to other methods in terms of mechanical properties [16]. Recent advancements, such as the use of rigid locking plates, have further improved the outcomes of arthrodesis by allowing earlier weight-bearing and reducing the risk of hardware failure [23]. These innovations highlight the evolving nature of surgical techniques and their potential to enhance outcomes for RA patients [24].

Future directions and innovations continue to evolve. Minimally invasive approaches such as arthroscopic arthrodesis are gaining attention [25]. These techniques offer potential advantages, including reduced surgical trauma, preservation of blood supply, and improved cosmetic outcomes [26]. While these methods are not yet widely adopted, they represent promising avenues for future research and clinical application [27].

In addition, the paradigm shift in RA treatment with advanced therapies has significant implications for surgical outcomes. Improved disease control may reduce the risk of postoperative complications and enhance the long-term success of reconstructive surgeries [20]. Further research is needed to explore the interaction between medical and surgical management in RA patients [22].

Beyond the techniques employed in this case, a variety of therapeutic strategies are available for addressing hallux varus in rheumatoid arthritis patients. These are summarized in Table 1.

### 4.4. Limitations

This case report presents certain limitations that warrant consideration. The lack of detailed information regarding the initial hallux valgus surgery inhibits a comprehensive understanding of the factors contributing to the iatrogenic deformity. Furthermore, the absence of a preoperative biomechanical analysis limits the capacity to objectively assess functional improvements.

As a single case study, the results are inherently patient-specific, are based on a subjective assessment by the authors, and may not be broadly generalizable to broader populations. Nonetheless, this report offers valuable insights into the management of hallux varus in the context of rheumatoid arthritis, emphasizing the complexities involved and identifying areas for future research.

Moreover, the CARE guidelines for clinical case reporting have been strictly adhered to, as outlined in Appendix A. These guidelines ensure a structured and transparent presentation of the case, facilitate reproducibility and enhance the report’s academic integrity. Our results are based solely on a subjective evaluation proposed by the authors.

## 5. Conclusions

Careful preoperative assessment and individualized surgical planning are essential for managing foot deformities in patients with rheumatoid arthritis (RA), with the goal of improving quality of life and preventing recurrence. Stabilized cases of RA are defined as those in which inflammatory activity is well controlled through pharmacological therapy or remission, thereby minimizing the risk of disease progression during the postoperative period. For these patients, joint-preserving techniques, such as the Scarf osteotomy, may be appropriate for correcting deformities without compromising joint function, particularly in mild to moderate cases.

In contrast, for patients with severely degenerated joint surfaces or advanced deformities, metatarsophalangeal (MTP) joint fusion remains the technique of choice. Fusion provides stability and pain relief and reduces the risk of deformity recurrence, particularly in cases with compromised bone quality. In the reported case, despite the observation of an asymptomatic bone non-union following MTP joint fusion surgery, the patient achieved significant improvements in functionality and pain relief. This outcome underscores the clinical success of MTP joint fusion in addressing hallux varus and associated symptoms, even in the presence of complications such as non-union.

Additionally, the combined use of resection arthroplasty for the lesser metatarsals and K-wire osteosynthesis offers effective alignment and stability, especially in patients with poor bone quality. This case highlights the importance of tailoring surgical decisions to the extent of joint damage, bone tissue quality, and the control of inflammatory disease. Meticulous surgical planning is critical to optimizing functional outcomes, restoring mobility, and enhancing patients’ quality of life.

While MTP joint fusion is often the preferred approach for advanced cases, alternative techniques such as resection arthroplasty or tendon transfer may be appropriate, depending on the specific clinical scenario. These options emphasize the necessity of a patient-centered approach to managing complex foot deformities associated with RA.

## Figures and Tables

**Figure 1 healthcare-13-00217-f001:**
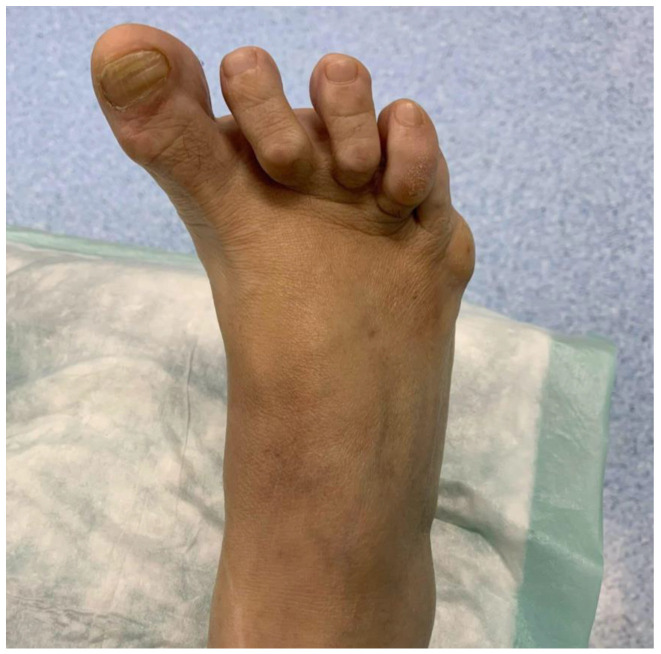
Image of the appearance of the right foot with preoperative hallux varus.

**Figure 2 healthcare-13-00217-f002:**
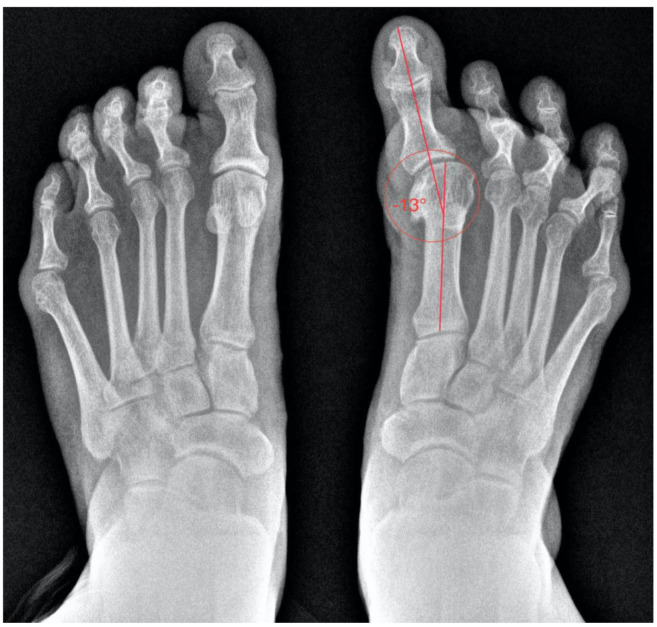
Preoperative anteroposterior (AP) weight-bearing radiograph of both feet demonstrating significant hallux varus deformity and associated structural abnormalities in the lesser toes.

**Figure 3 healthcare-13-00217-f003:**
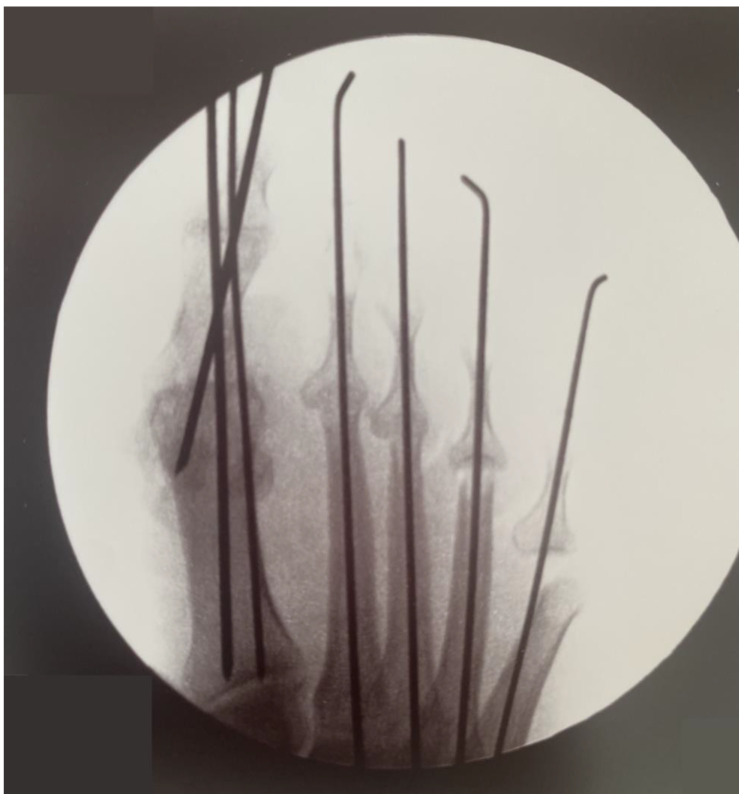
Postoperative AP radiograph of the right foot showing correction of hallux varus and lesser toe deformities with K-wires.

**Figure 4 healthcare-13-00217-f004:**
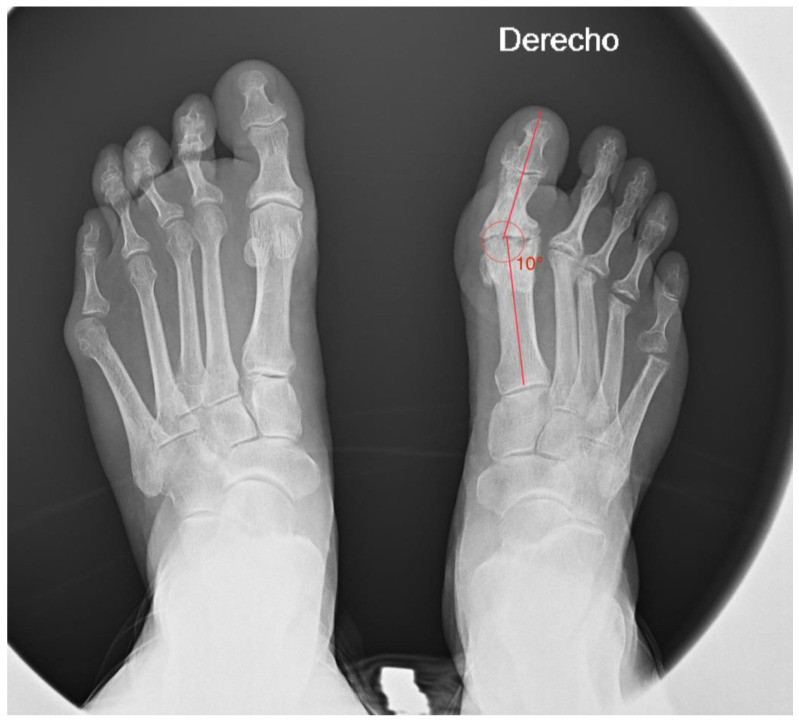
Image of the foot in load after 48 months.

**Table 1 healthcare-13-00217-t001:** Therapeutic Options for Managing Hallux Varus in Rheumatoid Arthritis Patients.

Main Authors (Refs)	Technique	Description	Advantages	Limitations
Matsumoto et al. [13,20]	Scarf Osteotomy	Osteotomy to correct mild-to-moderate deformities while preserving the joint.	Preserves joint, high satisfaction.	Less effective for severe deformities.
Horita et al. [15]	Resection Arthroplasty	Resection of the lesser metatarsal heads to relieve pain and correct toe deformities.	Reduces forefoot pressure, improves function.	Weakening of forefoot structure.
He et al. [18]				
He et al. [18,19]	MTP	Fusion of the metatarsophalangeal joint using K-wires or screws to stabilize and correct the deformity.	Stability, pain relief, prevents recurrence.	Risk of non-union, limited mobility.
Hyer et al. [23,24]	Arthrodesis			
Novak et al. [6]	Conservative Methods	Use of splints, functional taping, or orthotic insoles to manage symptoms and prevent progression.	Non-invasive, enhances patient comfort.	Limited efficacy in rigid deformities.
Li et al. [25,26]	Rigid Locking Plates	Advanced fixation plates for arthrodesis, improving stability and enabling early weight-bearing.	Greater stability, lower risk of mechanical failure.	Expensive, requires good bone quality.
Lui et al. [26]	Tendon Transfer	Transfer of the adductor hallucis or abductor hallucis tendon to rebalance forces and correct deformities.	Minimally invasive, preserves biomechanics.	Effective only in mild or early cases.
Schwagten et al. [27]				

## Data Availability

Data sharing is not applicable. No new data were created or analyzed in this study.

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
