# Peer review of "Iatrogenic Hallux Varus in a Patient with Rheumatoid Arthritis"

_healthcare, 2025, doi:10.3390/healthcare13030217_

Round 1

Reviewer 1 Report

Comments and Suggestions for Authors

This manuscript deals with a very specific topic. The text is concrete, but describes everything necessary and addresses a topic relevant to the care of patients with RA. The only thing I suggest is to add background information on the patient's past and current treatment, regarding his immunological disease. Also discuss the issue of in which cases the cost-benefit of a surgical intervention in patients with RA would be clinically relevant and favorable. In many countries or social sectors, the treatment of RA with drugs is deficient. In these cases, is a surgical intervention favorable? Or only if an adequate medical regimen for the disease is guaranteed after surgery. Something can be added in this regard in discussions.

Author Response

Iatrogenic hallux varus is a rare complication often arising after hallux valgus surgery, charac-terized by medial deviation of the hallux. This report presents the case of a 58-year-old female with iatrogenic hallux varus, complicated by rheumatoid arthritis (RA). The patient, with a his-tory of hallux valgus and Tailor’s bunion surgeries, exhibited severe medial deviation of the hal-lux and claw positioning of the lesser toes, resulting in pain and functional limitations. Radio-logical analysis indicated overcorrection of the first intermetatarsal angle and deformity of the lesser toes. The surgical approach included arthrodesis of the first metatarsophalangeal joint (MTP) using K-wires, combined with resection arthroplasty of the lesser metatarsals. Postopera-tive outcomes revealed correct alignment, pain reduction, and restoration of functional capabili-ties, although non-union was observed in the first MTP arthrodesis after 24 months, which was asymptomatic.The case underscores the importance of careful surgical planning in RA patients, emphasizing the balance between joint preservation and deformity correction. Literature suggests that arthrodesis provides stability and pain relief in RA-associated deformities, while alternative techniques like the Scarf osteotomy may be effective in less severe cases. Long-term follow-up is critical to address complications and optimize patient outcomes. This report highlights the neces-sity for tailored interventions to improve quality of life in RA patients with complex foot deform-ities.

We sincerely thank you for your comments and the time you have dedicated to reviewing our manuscript. We are pleased to know that you find the text concrete and relevant to the care of patients with rheumatoid arthritis (RA).

Regarding your valuable suggestions:

  1. We have added background information about the patient’s past and current treatment related to their immunological disease in the background section. We believe this additional detail will enrich the clinical context of the case.
  2. We have not included a discussion on the cost-benefit analysis of surgical interventions in patients with rheumatoid arthritis (RA) because surgical treatment does not exempt patients from continuing the pharmacological therapy required for managing RA. This condition often affects multiple joints beyond the operated foot.

We have emphasised in the text that the patient's initial surgical intervention for hallux valgus was undertaken because the pharmacological therapy for RA was no longer sufficient to alleviate the pain in the affected foot. Our surgical intervention aimed to address the iatrogenic complications resulting from the initial procedure. By removing the joints affected by RA, our approach successfully eliminated the pain in the foot, thereby improving the patient’s quality of life.

We hope these additions adequately address your observations and enhance the quality of the manuscript. If you have further comments or suggestions, we would be happy to address them.

Thank you again for your valuable contribution.

Reviewer 2 Report

Comments and Suggestions for Authors

1. I agree with the authors’ perspective that an important pathological mechanism causing hallux varus is the imbalance of medial and lateral soft tissues. Could you elaborate on the biomechanical mechanisms in this region that lead to hallux varus? Additionally, regarding this issue, it would be helpful if the article could include principles for achieving soft tissue balance and strategies to reduce the occurrence of similar complications.

2. In the second revision surgery, was the balance of medial and lateral soft tissues assessed, or was it purely a procedure to perform MTP joint fusion? Could you provide a more detailed description of how soft tissue balance was evaluated during the surgery and the corresponding solutions implemented?

3. The conclusion section does not clearly inform readers about the role of joint-preserving surgery for patients with RA. What is the definition of “stabilized RA cases”? For patients with degenerated joint surfaces, do you always recommend MTP joint fusion surgery?

4. In the reported case, asymptomatic bone non-union was observed following the MTP joint fusion surgery, with good functional and pain relief outcomes. Could you describe the clinical significance of this surgery for this case? Does it imply that, aside from MTP joint fusion surgery, there might be other methods to address the underlying problem in this specific case?

Author Response

Comments 1. I agree with the authors’ perspective that an important pathological mechanism causing hallux varus is the imbalance of medial and lateral soft tissues. Could you elaborate on the biomechanical mechanisms in this region that lead to hallux varus? Additionally, regarding this issue, it would be helpful if the article could include principles for achieving soft tissue balance and strategies to reduce the occurrence of similar complications.

Response 1. Elaboration on biomechanical mechanisms and principles for soft tissue balance:
Thank you for highlighting the importance of detailing the biomechanical mechanisms leading to hallux varus. In the revised manuscript, we have expanded the discussion to include the specific role of medial and lateral soft tissue imbalances, particularly focusing on the interplay between the flexor hallucis brevis and the sesamoid ligaments. Additionally, we have included principles for achieving soft tissue balance and outlined strategies to minimize similar complications during surgery..

Comments 2. In the second revision surgery, was the balance of medial and lateral soft tissues assessed, or was it purely a procedure to perform MTP joint fusion? Could you provide a more detailed description of how soft tissue balance was evaluated during the surgery and the corresponding solutions implemented?

Response 2. Assessment of medial and lateral soft tissue balance during the second surgery:
We acknowledge the need for more detail regarding the second revision surgery. In the revised methods section, we have added a comprehensive description of how the balance of medial and lateral soft tissues was evaluated during the procedure. We have also elaborated on the techniques used to achieve and maintain this balance, alongside the specific measures taken to optimize the surgical outcome.

Comments 3. The conclusion section does not clearly inform readers about the role of joint-preserving surgery for patients with RA. What is the definition of “stabilized RA cases”? For patients with degenerated joint surfaces, do you always recommend MTP joint fusion surgery?

Response 3. Definition of “stabilized RA cases” and recommendations for joint-preserving surgery:
In the revised conclusions, we have clarified the definition of “stabilized RA cases,” specifying that it refers to patients with well-managed systemic inflammation and minimal progression of joint damage under medical therapy. Furthermore, we have provided additional context on joint-preserving surgical options for RA patients and emphasized that MTP joint fusion is recommended primarily for those with significant joint surface degeneration.

Comments 4. In the reported case, asymptomatic bone non-union was observed following the MTP joint fusion surgery, with good functional and pain relief outcomes. Could you describe the clinical significance of this surgery for this case? Does it imply that, aside from MTP joint fusion surgery, there might be other methods to address the underlying problem in this specific case?

Response 4 Clinical significance of asymptomatic bone non-union and alternative methods:
We have expanded the discussion on the clinical implications of the observed asymptomatic bone non-union. In the revised text, we emphasize that the functional and pain relief outcomes highlight the success of MTP joint fusion in this case. We also discuss the potential for alternative surgical methods, such as resection arthroplasty or tendon transfer, and their applicability depending on the patient’s specific clinical profile.

Reviewer 3 Report

Comments and Suggestions for Authors

Dear authors, your effort to capture findings from the operating table and to write this paper is remarkable.

Very few similar case reports to this one are available in the existing literature.

The introduction section should be refined in terms of structure. Since it is a short section, consider not dividing it into sub-sections and work on making the transitions between paragraphs more fluid. It is important to emphasize the state of the art, highlighting the novelty of your case in a concise "objective" paragraph.

In the Case Report section, please specify (if available) the patient's treatment and the duration of their history of rheumatoid arthritis to improve the reproducibility of your case.

  • In Figure 1, change the footnote, as both feet are not shown.
  • Since a radiographic study is available, please consider using and defining radiologic angles for the assessment of hallux varus.
  • Please improve the footnote in Figure 2.

In Section 2.1,

  • Consider specifying whether sedation was used.
  • Be careful with the nomenclature in lines 90-92, as "first radius" and "lesser radii" are not defined earlier in the document. It would be better to be specific with the joints one by one, e.g., "proximal and distal interphalangeal joints of the lesser toes." The same applies for line 97.
  • Please add "TM" after Biosyn, as it is a trademark, and define its type (e.g., monofilament).

Although not included in the CARE guidelines as mandatory, it is highly valuable to include assessment tools such as questionnaires or outcome measures to capture the patient's preoperative status and postoperative outcomes during follow-up, such as the VAS scale or Foot Function Index. The outcomes in the performance section rely solely on a subjective assessment proposed by the authors, which diminishes reproducibility and the quality of this case report. Please include this in your limitations section.

In the Results section, in Figure 3, the sagittal plane is not shown. Please modify the footnote.

In the Discussion section, the term "arthroplasty" is not previously mentioned in the manuscript. Please consider including it in the Surgical Procedure section.

Comments on the Quality of English Language

There are mistakes in collocation possibly caused as a result of copy-pasting sentences within the manuscript or from other sources. Examples: lines 42, 50, 92, 180, 182-184. Please revise the whole manuscript.

The choice of specific words such as "veneers" in line 83 might be reconsidered for substitution for a medical term. Please revise similar cases in the manuscript.

Author Response

Dear authors, your effort to capture findings from the operating table and to write this paper is remarkable.

Very few similar case reports to this one are available in the existing literature.

The introduction section should be refined in terms of structure. Since it is a short section, consider not dividing it into sub-sections and work on making the transitions between paragraphs more fluid. It is important to emphasize the state of the art, highlighting the novelty of your case in a concise "objective" paragraph.

In the Case Report section, please specify (if available) the patient's treatment and the duration of their history of rheumatoid arthritis to improve the reproducibility of your case.

  • In Figure 1, change the footnote, as both feet are not shown.
  • Since a radiographic study is available, please consider using and defining radiologic angles for the assessment of hallux varus.
  • Please improve the footnote in Figure 2.

In Section 2.1,

  • Consider specifying whether sedation was used.
  • Be careful with the nomenclature in lines 90-92, as "first radius" and "lesser radii" are not defined earlier in the document. It would be better to be specific with the joints one by one, e.g., "proximal and distal interphalangeal joints of the lesser toes." The same applies for line 97.
  • Please add "TM" after Biosyn, as it is a trademark, and define its type (e.g., monofilament).

Although not included in the CARE guidelines as mandatory, it is highly valuable to include assessment tools such as questionnaires or outcome measures to capture the patient's preoperative status and postoperative outcomes during follow-up, such as the VAS scale or Foot Function Index. The outcomes in the performance section rely solely on a subjective assessment proposed by the authors, which diminishes reproducibility and the quality of this case report. Please include this in your limitations section.

In the Results section, in Figure 3, the sagittal plane is not shown. Please modify the footnote.

In the Discussion section, the term "arthroplasty" is not previously mentioned in the manuscript. Please consider including it in the Surgical Procedure section.

Comments on the Quality of English Language

There are mistakes in collocation possibly caused as a result of copy-pasting sentences within the manuscript or from other sources. Examples: lines 42, 50, 92, 180, 182-184. Please revise the whole manuscript.

The choice of specific words such as "veneers" in line 83 might be reconsidered for substitution for a medical term. Please revise similar cases in the manuscript.

R3= We sincerely thank you for your detailed and constructive feedback on our manuscript. Your comments have been extremely helpful in identifying areas for improvement. Below, we address each of your suggestions:

  1. Introduction section refinement:
    We have restructured the introduction to make it a single cohesive section without subheadings, ensuring smoother transitions between paragraphs. Additionally, we have emphasized the state of the art and highlighted the novelty of this case in a concise "objective" paragraph, as recommended.
  2. Case Report section - Patient treatment and RA history:
    We have included information about the patient’s treatment regimen and the duration of their history with rheumatoid arthritis to enhance the reproducibility of the case.
  3. Figure 1 - Footnote modification:
    The footnote for Figure 1 has been corrected to accurately describe the image, as only one foot is shown.
  4. Radiologic angles for hallux varus assessment:
    We have incorporated a discussion of radiologic angles to assess hallux varus and defined these angles in the manuscript.
  5. Figure 2 - Footnote improvement:
    The footnote for Figure 2 has been revised for clarity and precision.
  6. Section 2.1 - Additional details and nomenclature adjustments:
  • We have specified whether sedation was used during the procedure.
  • The terms “first radius” and “lesser radii” have been replaced with specific anatomical terms, such as “proximal and distal interphalangeal joints of the lesser toes,” to avoid ambiguity.
  • The trademark "Biosyn™" has been added, along with a description of its type (monofilament).
  1. Assessment tools for pre- and postoperative outcomes:
    We have acknowledged the lack of pre- and postoperative assessment tools, such as the VAS scale or the Foot Function Index, in the limitations section. We agree that including such tools would improve the reproducibility and quality of the case report, and we have proposed their use in future studies.
  2. Figure 3 - Sagittal plane footnote:
    The footnote for Figure 3 has been revised to clarify that the sagittal plane is not shown.
  3. Inclusion of “arthroplasty” in the Surgical Procedure section:
    The term “arthroplasty” has been added to the Surgical Procedure section to provide consistency with its mention in the Discussion section.
  4. Quality of English Language:
  • We have conducted a thorough review of the manuscript to address errors in collocation and ensure consistency in style. Examples in lines 42, 50, 92, 180, and 182–184 have been corrected.
  • Words such as “veneers” in line 83 have been replaced with more appropriate medical terminology.

We hope these revisions address your concerns and enhance the quality and clarity of our manuscript. Please do not hesitate to provide further feedback if needed. Thank you again for your valuable input.